# Mechanisms and Clinical Implications of Human Gut Microbiota-Drug Interactions in the Precision Medicine Era

**DOI:** 10.3390/biomedicines12010194

**Published:** 2024-01-16

**Authors:** Shuaiqi Wang, Dianwen Ju, Xian Zeng

**Affiliations:** Department of Biological Medicines & Shanghai Engineering Research Center of Immunotherapeutics, School of Pharmacy, Fudan University, Shanghai 201203, China; wangsq20@fudan.edu.cn

**Keywords:** gut microbiota, precision medicine, microbiota-drug interaction, drug metabolism

## Abstract

The human gut microbiota, comprising trillions of microorganisms residing in the gastrointestinal tract, has emerged as a pivotal player in modulating various aspects of human health and disease. Recent research has shed light on the intricate relationship between the gut microbiota and pharmaceuticals, uncovering profound implications for drug metabolism, efficacy, and safety. This review depicted the landscape of molecular mechanisms and clinical implications of dynamic human gut Microbiota-Drug Interactions (MDI), with an emphasis on the impact of MDI on drug responses and individual variations. This review also discussed the therapeutic potential of modulating the gut microbiota or harnessing its metabolic capabilities to optimize clinical treatments and advance personalized medicine, as well as the challenges and future directions in this emerging field.

## 1. Introduction

The human gut microbiota constitutes a highly intricate and dynamically evolving ecosystem housing an abundance of microorganisms, encompassing bacteria, archaea, fungi, viruses, and protozoa [1]. These microorganisms fulfill indispensable roles across diverse facets of human health and maladies, including digestion, immunity, metabolism, and behavior [2]. Furthermore, the gut microbiota engages in intricate interactions with a myriad of exogenous substances, including dietary constituents, environmental contaminants, and pharmaceutical agents [3]. These interactions hold profound implications for the pharmacokinetics and pharmacodynamics of pharmaceuticals, as well as their overall efficacy and safety profiles [4].

The nexus between the human gut microbiota and pharmaceuticals stands as a swiftly burgeoning domain known as pharmacomicrobiomics [5]. It seeks to unveil the molecular mechanisms underpinning the inter-individual clinical ramifications arising from gut microbiota-mediated drug metabolism, alongside exploring the reciprocal impacts of pharmaceutical agents on the composition and functionality of the gut microbiome [6]. A plethora of studies have unveiled that the gut microbiota exerts regulatory control over the bioavailability, biotransformation, distribution, elimination, and potential toxicity of a diverse array of pharmaceuticals, spanning antibiotics, antihypertensives, antidiabetics, anticancer agents, antipsychotics, and analgesics [7,8]. Furthermore, pharmaceutical agents have the potential to perturb the diversity and stability of the gut microbiome, consequently, precipitating dysbiosis, diminished colonization resistance, heightened susceptibility to infections, and alterations in host metabolism [9,10,11].

The dynamic interplay between the human gut microbiota and pharmaceuticals holds profound implications for individualized drug responses and personalized medicine [12,13,14]. On one facet, it serves to elucidate the inter-individual variability in drug efficacy, susceptibility to drug-induced adverse reactions, and proclivity to drug-drug interactions [15]. In addition, it represents innovative avenues for optimizing drug regimens by modulating the gut microbiome or harnessing its metabolic capabilities. These approaches encompass the deployment of probiotics, prebiotics, synbiotics, fecal microbiota transplantation, or microbial enzyme inhibitors [16]. Thus, comprehending the integral role of the gut microbiome in drug disposition and therapeutic action holds the potential to improve drug treatments and enhance patient care.

In this review, we attempted to undertake an in-depth examination of the current landscape of the interplay between the human gut microbiota and pharmaceuticals, the molecular mechanisms and clinical implications pertaining to gut microbiota-mediated drug metabolism and the pharmaceutical-induced alterations in the microbiome, and the challenges and opportunities inherent in the development of novel strategies for modulating gut microbiota-drug interactions, with an eye toward advancing the realm of personalized medicine.

## 2. Molecular Mechanisms of Microbiota-Drug Interactions

The gut microbiome engages in multifaceted interactions with drugs. These interactions carry significant ramifications for multiple facets, encompassing pharmacokinetics, pharmacodynamics, efficacy, and toxicity of the bioactive agents, alongside influencing the composition, functionality, and homeostasis of the gut microbiome [17]. A comprehensive comprehension of the molecular mechanisms underlying these interactions is imperative to optimize the utilization of bioactive compounds and leverage the potential of the gut microbiome as a therapeutic target or modulator. The effects of drugs on the abundance of individual microbes and dysbacteriosis have been reviewed thoroughly elsewhere [18,19,20,21], therefore, this review will emphasize the mechanisms and clinical implications of MDI on drugs, which can be roughly divided into three types: direct transformation of drugs by microbiota, indirect transformation of drugs by microbiota, and bioaccumulation of drugs by microbiota, as illustrated in the Figure 1.

### 2.1. Direct Metabolic Transformation of Drugs by Microbiota

The influences on drug biotransformation through two primary mechanisms, direct microbial transformations and indirect modulation of drug-host metabolizing enzymes and transporters [1].

Gut microbes metabolize drugs through oxidation and reduction reactions, which may lead to processes of activation, inactivation, or toxicity. A frequently observed direct metabolic transformation involves the reduction in azo and nitro groups, commonly found in prodrugs designed for activation within the colon. For instance, sulfasalazine, a prodrug used in the treatment of inflammatory bowel disease (IBD), undergoes reduction by the gut microbiota, liberating 5-aminosalicylic acid (5-ASA), the biologically active anti-inflammatory component [22]. Similarly, balsalazide and olsalazine are prodrugs relying on gut microbiota-mediated reduction to release 5-ASA [23]. Metronidazole, an antibiotic featuring a nitro group, also needs gut microbiota to be reduced to its toxic radical [24]. However, it is important to note that azo and nitro group reduction can also result in drug inactivation or the formation of toxic metabolites. For example, some nitroaromatic compounds such as dinitrotoluene and trinitrotoluene, can be reduced by the gut microbiota, yielding carcinogenic aromatic amines.

Another mode of direct metabolic transformation is the hydrolysis of ester and amide bonds, which may either activate or inactivate drugs. Lovastatin, a cholesterol-lowering medication, undergoes hydrolysis by the gut microbiota, yielding lovastatin acid, characterized by enhanced bioavailability and potency compared to the parent compound [25]. In addition, oseltamivir, an antiviral agent targeting influenza virus neuraminidase, represents an ester prodrug subject to hydrolysis by both gut microbiota and hepatic carboxylesterases, leading to the formation of active oseltamivir carboxylate [26]. Conversely, acarbose, an antidiabetic drug inhibiting intestinal alpha-glucosidases, is hydrolyzed by the gut microbiota, generating inactive metabolites [27].

A third aspect of direct drug metabolism involves the deconjugation of glucuronide and sulfate conjugates, which is a special form of hydrolysis. Deconjugation of these conjugates can facilitate the reabsorption of drugs or their metabolites into the systemic circulation, consequently prolonging their exposure and effects [28,29]. For instance, morphine-6-glucuronide (M6G) undergoes deconjugation by the gut microbiota, liberating free morphine, which can be reabsorbed and contribute to the analgesic and addictive properties of morphine [30]. Similarly, inactive estrogen conjugates of glucuronides or sulfates can be deconjugated by the gut microbiota, yielding free estrone that may be reabsorbed and impact estrogen-dependent processes [31].

### 2.2. Indirect Impact of Microbiota on Drug Metabolism

The gut microbiota indirectly influences drug biotransformation through various mechanisms. One such mechanism involves the modulation of host drug-metabolizing enzymes and transporters. This modulation is achieved by the production of a range of metabolites, including short-chain fatty acids (SCFAs), indoles, and secondary bile acids. Specific gut bacteria have the capacity to produce SCFAs, such as acetate, propionate, and butyrate. These SCFAs can modulate the expression and activity of hepatic cytochrome P450 (CYP) enzymes through various mechanisms, such as activation of peroxisome proliferator-activated receptor (PPAR), inhibition of histone deacetylases (HDACs), or alteration of redox status [32]. Similarly, certain gut bacteria can generate indole derivatives through tryptophan metabolism, which can either inhibit CYP enzymes or induce phase II conjugation reactions [33,34].

Another indirect pathway through which the gut microbiota affects drug metabolism is by influencing the enterohepatic circulation of drugs. Enterohepatic circulation is a process in which drugs and xenobiotics undergo metabolism in the liver, are secreted into the bile, reabsorbed in the intestine, and subsequently returned to the liver via portal blood. In this process, the gut microbiota plays a crucial role in bile acid metabolism by affecting lipid and lipid-soluble vitamin absorption and maintaining intestinal barrier function [35]. In addition, gut microbes can also impact drugs undergoing enterohepatic recycling by altering host enzyme metabolic processes, potentially increasing exposure to toxic metabolites [5].

### 2.3. Bioaccumulation of Drugs in Microbiota

Apart from drug biotransformation, another significant aspect of drug-microbiota interaction is drug bioaccumulation. Bioaccumulation refers to the process wherein certain drugs are absorbed and retained by the bacteria inhabiting the human gut. The primary mechanism behind bioaccumulation involves the binding of drugs to bacterial cell walls, membranes, or intracellular components, thereby impeding their absorption or metabolism by the host. Additionally, some drugs can be actively or passively transported into bacterial cells [1].

Bioaccumulation has far-reaching consequences, influencing drug availability and bacterial metabolism. These effects extend to microbiota composition, pharmacokinetics, side effects, and drug responses [36,37]. On the one hand, drugs bioaccumulated by bacteria may result in lower plasma concentrations and reduced therapeutic efficacy within the host [38]. One example is the antidepressant Duloxetine, which was found to be bioaccumulated by gut bacteria, leading to reduced host blood drug concentration and decreased therapeutic effect [36]. On the other hand, when the bacteria die and release the accumulated drug into the intestinal lumen, it may lead to toxicity and overdose symptoms, such as increased blood pressure, vomiting, and cardiac arrhythmia. Furthermore, drugs bioaccumulated by bacteria can selectively impact the growth and survival of different bacterial species within the gut and increase the risk of antibiotic resistance. For instance, tetracycline can select resistant bacteria and transfer antibiotic resistance genes to other bacteria, leading to the development of antibiotic resistance in gut microbiota [39]. It can alter the diversity and stability of the gut microbiota, affecting its metabolic and immunological functions.

## 3. Impact of MDI on Drug Responses

There are bidirectional interactions between drugs and gut microbiota, as drugs can shape the composition of gut microorganisms, while the diverse gut microbiota can impact drug efficacy and toxicity in turn [40]. In recent years, the intricate interactions between gut microbes and commonly used non-antibiotic drugs have attracted considerable attention. Figure 2 illustrates examples of the impact of gut microbiota (species) on drug efficacy and toxicity (the detailed interactions and corresponding references were summarized in Appendix A). In addition, it has also been demonstrated that microbiota is able to directly or indirectly modulate our immune status to affect various clinical settings such as cancer immunotherapy and COVID-19 management [41,42]. Those findings provide us with a comprehensive understanding of how the microbiome metabolizes drugs and affects treatment outcomes, which holds the potential to enable the manipulation of the gut microbiome to enhance treatment effectiveness [4].

### 3.1. Efficacy

Microbial metabolism plays a pivotal role in modulating drug efficacy by affecting the concentration of active compounds. The gut microbiota’s impact on drug efficacy hinges on its ability to convert prodrugs into their active forms or transform active drugs into inactive forms.

On one hand, the gut microbiota can augment drug efficacy by converting prodrugs into their active forms. For instance, bacterial azoreductases in the colon facilitate the conversion of sulfasalazine into its active forms [43]. On the other hand, microbial metabolism can also diminish drug efficacy by converting active drugs into inactive forms. For instance, *Eggerthella* can metabolize digoxin into inactive forms [44]. Additionally, microbial metabolism can increase or decrease drug bioavailability by interfering with efflux transporters in the gut epithelium. A notable example is digoxin with poor absorption due to rapid efflux by P-glycoprotein (P-gp) transporters [45]. However, gut bacteria can counteract this by regulating P-gp in the intestinal epithelium [46].

### 3.2. Toxicity

Microbial metabolism is a complex process involving the breakdown of various compounds, including drugs. During this process, certain microbes can produce toxic metabolites that contribute to drug toxicity. Gut microbiota-produced metabolizing enzymes can generate toxic intermediates, such as trimethylamine N-oxide (TMAO) [47]. These toxic metabolites have the potential to elicit adverse reactions in humans, including liver damage, kidney failure, and even mortality [48]. Furthermore, some gut bacteria can catalyze the reduction in the nitro group in metronidazole, generating a cytotoxic metabolite with the potential to induce DNA damage and mutagenesis [33].

In addition to generating toxic metabolites, microbial metabolism can increase drug exposure. The gut microbiota can metabolize and modify the chemical structure of numerous orally administered xenobiotics, encompassing environmental pollutants, dietary components, and therapeutic drugs. This amplifies the enzymatic repertoire within the gut, surpassing that in the liver, thereby affecting the metabolism and pharmacological effects of numerous drugs, either directly or indirectly [1]. Furthermore, gut microbes influence drugs undergoing enterohepatic recycling by altering host enzyme metabolic processes, thereby enhancing exposure to toxic metabolites [5].

### 3.3. Immune Modulation

The gut microbiota plays a multifaceted role in various physiological processes, encompassing metabolism, inflammation, immunity, and neurology, as supported by numerous references [49,50,51]. Notably, the gut microbiota and the immune system have co-evolved, fostering a mutually beneficial relationship where both entities exert reciprocal influence [52,53].

In terms of immunity modulation, the gut microbiota wields the capacity to impact both innate and adaptive immune responses. It extends its influence to antitumor immune responses within the tumor microenvironment (TME), engaging both innate and adaptive immune cell populations [54]. A primary mechanism through which the gut microbiota exerts its influence on antitumor immunity involves the production of metabolites. These small molecules can traverse from the gut to TME and influence the activation and function of immune cells, which may contribute to an enhanced antitumor immune response [51]. The intricate interplays between gut microbiota-derived metabolites, local and systemic immune responses, and the modulation of the TME collectively underscores the multifaceted impacts of the gut microbiota on cancer immunotherapies.

Furthermore, the gut microbiota is also involved in immunosuppression and transplantation medicine. Immunosuppressive drugs have the potential to modify the composition of the gut microbiota, subsequently impacting both drug metabolism and the immune system of transplantation recipients. For instance, mycophenolic acid (MPA), an immunosuppressive agent inhibiting T and B lymphocyte proliferation associated with organ rejection [55], has been found to induce gut microbiota dysbiosis [56]. The dysbiosis not only disrupts the enterohepatic recirculation (EHR) of MPA, which leads to alterations of pharmacokinetic and pharmacodynamic profiles of MPA [57], but also elevates the risk of infections, inflammation, and graft-versus-host disease (GvHD) in transplant patients [58,59]. Strategies aimed at modulating the gut microbiota, such as employing probiotics, prebiotics, or fecal microbiota transplantation (FMT), represent potential avenues to improve clinical outcomes of transplantation [60]. These interventions hold promise for mitigating dysbiosis-associated complications, thereby optimizing the efficacy and safety of immunosuppressive therapies in precision medicine in transplantation.

Recent research findings emphasize the pivotal role played by the composition and function of an individual’s gut microbiota in shaping immune responses to vaccinations [61]. This interaction between the host and gut microbes is posited to be a central determinant of these immune responses [52]. Both clinical cases and animal models have demonstrated the ability of the gut microbial community to influence the effectiveness of vaccines [62]. This emerging knowledge holds the potential to be harnessed for the enhancement of therapeutic vaccination strategies through deliberate manipulation of the gut microbiota.

### 3.4. Impact on COVID-19

COVID-19 is a respiratory disease that can lead to severe pneumonia, acute respiratory distress syndrome (ARDS), multiorgan failure, and death in some cases. The main target of COVID-19 is the angiotensin-converting enzyme 2 (ACE2) receptor, which is highly expressed in the alveolar epithelial cells of the lungs. Interestingly, ACE2 is also present in the enterocytes of the small intestine, indicating that COVID-19 can invade and replicate in the gut [63]. Studies have demonstrated that COVID-19 patients have diminished bacterial diversity and richness, as well as altered microbial profiles. These changes in gut microbiota remain even after COVID-19 recovery, implying a potential connection with long-term complications or post-acute COVID-19 syndrome (PACS) [64].

The efficacy and safety of COVID-19 vaccines may be affected by gut microbiota, as it has been proven to regulate the antibody response to various vaccines, such as influenza11, polio, rotavirus, and oral cholera vaccines [64,65,66]. The possible mechanisms by which gut microbiota may impact vaccine immunization include increasing type I interferon production, triggering antiviral immunoglobulin A secretion, enhancing T cell activation and differentiation, and supporting regulatory T cell function [67,68]. A recent study has revealed that gut microbiota composition is correlated with SARS-CoV-2 vaccine immunogenicity and adverse events in adults. The study discovered that *Bifidobacterium adolescentis* was consistently higher in subjects with high neutralizing antibodies to vaccines [69]. It also detected that *Prevotella copri* and two *Megamonas* species were more prevalent in individuals with fewer adverse events after the vaccines, suggesting that these bacteria may have an anti-inflammatory role in the host immune response.

Moreover, gut microbiota modulation by probiotics, prebiotics, synbiotics, fecal microbiota transplantation (FMT), or personalized nutrition may be potential strategies to prevent or treat COVID-19 and its complications. Studies have indicated that these strategies may have positive effects on COVID-19 patients or high-risk groups by relieving symptoms, accelerating viral clearance, lowering disease severity, avoiding complications, or boosting vaccine response [64,70].

### 3.5. Interindividual Variability

Achieving consistent therapeutic outcomes poses a significant challenge in drug development and clinical practice, partially attributable to interindividual variability in gut microbiota composition and function. This variability leads to divergent drug responses among individuals, making it challenging to predict drug efficacy and toxicity [71]. On one hand, the gut microbiota can modulate drug metabolism and pharmacokinetics by influencing the expression and activity of drug-metabolizing enzymes and transporters [72,73]. On the other hand, the gut microbiota can impact drug absorption, distribution, and elimination by interacting with the host immune system and altering intestinal barrier function [74]. Notably, the gut microbiota’s influence on the efficacy and toxicity of anticancer drugs [75] and its role in shaping responses to cancer immunotherapy are highly correlated with interindividual variability [76]. Moreover, the gut microbiota can influence responses to dietary interventions aimed at enhancing host outcomes and reducing disease risk [75].

In conclusion, the variability in gut microbiota composition and function among individuals contributes to significant disparities in drug responses. This relationship between gut microbiota composition and interindividual variability in drug responses represents an emerging field of research with substantial implications for drug development and clinical practice. Further investigations are essential to elucidate the underlying mechanisms and develop strategies to optimize drug therapy based on individual gut microbiota profiles.

## 4. Clinical Implications of MDI

The gut microbiota’s pivotal role in the pathogenesis and progression of various diseases is increasingly acknowledged in contemporary research. Consequently, it emerges as a promising target for pharmacological intervention [77]. In Table 1, we have compiled information from 32 completed clinical trials with outcomes pertinent to the involvement of gut microbiota in treatments from the ClinicalTrials.gov database. These trials encompass a range of conditions, including inflammatory bowel disease (IBD), irritable bowel syndrome (IBS), non-alcoholic fatty liver disease (NAFLD), *Helicobacter pylori* infection, cancers and other symptoms. These trials collectively underscore the potential of gut microbiota modulation in disease management. Nevertheless, further investigations are imperative to validate the long-term efficacy, optimal dosages, timing, frequency, duration, and method of administration for these interventions.

In summary, the implications of the gut microbiota in drug treatments have unveiled new avenues for therapeutic strategies. These strategies encompass the manipulation of the gut microbiota to enhance or diminish drug metabolism, the design of drugs with reduced susceptibility to gut microbiota-mediated metabolism, the development of probiotics or prebiotics capable of modulating the gut microbiota’s composition or function, and the implementation of personalized medicine approaches that account for individual variations in the gut microbiota [97,98,99].

### 4.1. Modulation of Microbiota by Probiotics and Prebiotics

One effective strategy for modulating the gut microbiota involves the use of probiotics or prebiotics. Probiotics are live microorganisms that, when consumed in adequate quantities, confer health benefits. Prebiotics, on the other hand, are non-digestible food components that selectively stimulate the growth of beneficial gut bacteria [100]. Probiotics and prebiotics can influence the gut microbiota by increasing its diversity, enhancing its stability, restoring its balance, inhibiting the growth of pathogens, producing metabolites, and modulating immune responses [101].

Numerous studies have demonstrated that probiotics and prebiotics have a positive impact on overall health by preventing pathogen invasion and reducing the risk of conditions such as obesity, type 2 diabetes, inflammatory bowel disease, cancer, cardiovascular diseases, liver disorders, and central nervous system disorders [97,102]. For example, in the context of inflammatory bowel disease (IBD), probiotics, prebiotics, or synbiotics (combinations of both) can induce or maintain remission, lower the disease activity index, and prevent relapses [100]. Probiotics can also increase the abundance of beneficial bacteria, such as *Bifidobacteria parabacteroides*, *Alloprevotella*, and *Alistipes* [103]. In cancer treatment, probiotics or prebiotics can enhance the effectiveness of chemotherapy, immunotherapy, or radiotherapy by modulating the gut microbiota and the immune system [4]. Furthermore, they can alleviate some of the side effects associated with cancer treatment, such as diarrhea, mucositis, infections, and inflammation. In the case of diabetes, probiotics or prebiotics can improve glycemic control, enhance insulin sensitivity, and positively affect lipid profiles by influencing the gut microbiota and its metabolites, thus reducing the risk of diabetic complications, including nephropathy, neuropathy, and cardiovascular diseases [104].

In addition to probiotics and prebiotics, post-biotics such as bioactive metabolites produced by the gut microbiota and cell-wall components released by probiotics have been demonstrated to inhibit pathogen growth, maintain microbiota balance, and regulate immune responses [97]. However, it is essential to note that while probiotics can offer health benefits, they may also lead to side effects such as bloating and increased thirst [105,106]. Therefore, conducting additional clinical trials is necessary to assess the safety and efficacy of specific probiotics or prebiotics for various drugs and diseases.

### 4.2. Pharmacomicrobiomics

Inter-individual variability in drug response represents a significant challenge in clinical practice. Pharmacomicrobiomics is a burgeoning field of research that explores the intricate interplay between the gut microbiome, the host, and drugs. Its primary objective is to elucidate how the gut microbiome influences drug responses and disposition, encompassing processes such as drug absorption, distribution, metabolism, and excretion [107]. The gut microbiome is estimated to house over 100 trillion prokaryotes, representing more than 1000 distinct species [108]. These microorganisms can impact drug responses by producing enzymes that modify drug molecules, altering the expression and activity of host enzymes involved in drug metabolism, and modulating immune responses and inflammation [109]. The composition of the gut microbiome varies significantly among individuals, depending on factors such as diet, lifestyle, genetics, and environmental exposures [110]. This variability can lead to divergent drug outcomes among patients, including variations in efficacy, side effects, and drug interactions [111].

Understanding the role of the gut microbiome in drug metabolism may facilitate the elucidation of drug response variations [17]. For instance, microbial communities can be readily manipulated through various strategies, including FMT, probiotics, and antibiotics [112]. This presents opportunities for developing innovative therapeutics and personalized drug treatments that are not influenced or activated by microbiome-mediated processes [113]. Moreover, certain bacterial strains may possess the capacity to modulate cancer progression and responses to therapy, thereby enhancing the prospects of precision medicine tailored to microbiota-related considerations in both treatment and prognosis [76].

Profiling the microbial composition and function of each patient using metagenomic sequencing and metabolomics based on gut microbiome analysis holds promise for personalized medicine [114]. This approach can shed light on how the microbiome affects the pharmacokinetics (absorption, distribution, metabolism, and excretion) and pharmacodynamics (mechanism of action and therapeutic effects) of specific drugs [115]. For instance, metagenomic sequencing can identify microbial genes that encode enzymes capable of activating or inactivating prodrugs (inactive precursors that are converted to active drugs in the body) [37]. Meanwhile, metabolomics can quantify the levels of drug metabolites produced by the microbiome in the blood or urine [116]. These data can be integrated with other omics data, including genomics, proteomics, and transcriptomics, to provide a comprehensive understanding of how the host and the microbiome interact with drugs.

### 4.3. Insights for Rational Drug Discovery

Prodrugs are pharmacologically inert compounds that require biotransformation prior to manifesting their pharmacological effects [117]. The gut microbiota can metabolize prodrugs, which are inactive or less active compounds that need to be converted into active drugs in the body [118]. This metabolic process can have significant implications for drug development, as it can influence the bioavailability, efficacy, toxicity, and inter-individual variability of the prodrugs. The metabolic process can be direct or indirect, depending on whether the prodrug is transformed by the gut bacteria or by the host enzymes affected by the gut bacteria [23]. Therefore, to optimize the design and delivery of prodrugs, and to improve their therapeutic outcomes, it is important to understand how various factors affect the gut microbial metabolism of prodrugs [119,120]. These factors include the composition and diversity of the gut microbiota, the diet and lifestyle of the host, the drug dosage and formulation, and the co-administration of other drugs or probiotics. Table 2 shows typical examples of prodrugs that are metabolized by gut microbiota to produce active compounds.

In recent years, microbiome-based therapeutics have also witnessed significant advancements. These therapeutic approaches aim to manipulate the gut microbiome through additive, subtractive, or modulatory interventions utilizing native or engineered microbes, antibiotics, bacteriophages, and bacteriocins. Among the microbiome-based therapeutic strategies currently under development, probiotic therapies are the most advanced [136]. This innovative strategy holds the potential to surmount the limitations associated with conventional therapeutics by providing personalized, harmonized, dependable, and sustainable treatment modalities [137]. Empirical evidence has demonstrated that targeting and manipulating the microbiome can ameliorate human health and address various medical conditions by restoring a healthy balance of bacterial populations [138].

In summary, microbial metabolism constitutes a pivotal factor that can exert profound influence over the development of pharmaceuticals. The exploration of how the gut microbiome impacts the metabolism, mechanism of action, and toxicity of drugs in the human body offers a promising avenue for optimizing prodrugs and microbiome-specific therapeutic interventions tailored to the diverse needs of individual patients and various medical conditions.

## 5. Challenges and Future Directions

The dynamic interaction between gut bacteria and pharmaceutical agents constitutes a rapidly evolving realm of scientific inquiry with profound implications for the realm of personalized medicine. Nevertheless, this burgeoning field confronts several challenges: The foremost challenge pertains to standardization. The establishment of meticulously standardized protocols for investigating interactions between the gut microbiota and drugs assumes paramount importance in ensuring consistent research outcomes. This necessitates the standardization of methodologies encompassing sample collection, storage, and processing, as well as the rigorous application of uniform procedures for data analysis and interpretation [23]. A second pivotal issue lies in the ethical dimension. The deliberate manipulation of the gut microbiota for therapeutic purposes engenders a complex web of ethical concerns [139]. These concerns encompass issues pertaining to privacy, informed consent, and the potential for the improper use of personal health information [140]. Additionally, the realm of regulatory considerations merits attention, with a view to ensuring the safety and efficacy of interventions grounded in the microbiome [141,142]. Furthermore, the translation of microbiome-based interventions into clinical practice mandates an expansive research agenda and rigorous validation procedures. This endeavor encompasses the imperative task of demonstrating the efficacy of such interventions in large-scale clinical trials, unraveling their intricate mechanisms of action, and scrutinizing potential side effects [143,144].

While studying the interplay between gut bacteria and drugs presents numerous challenges, it also offers several promising avenues for future research within this domain. It is imperative that forthcoming investigations prioritize the comprehension of the intricate mechanisms governing interactions between gut bacteria and drugs, the innovation of novel techniques for modulating the gut microbiota, and the exploration of potential therapeutic applications stemming from these interactions [76]. Tackling these obstacles holds the potential to enhance our comprehension of MDI and facilitate the integration of microbiome-based/targeted interventions into clinical practice.

## 6. Conclusions

This manuscript attempted to depict the landscape of the molecular mechanisms and clinical implications of MDI by providing an overview of the current state of knowledge concerning the intricate relationship between human gut microbiota and pharmaceutical agents. We emphasized the molecular mechanisms underpinning this interaction, its implications for drug response and toxicity variations, and the potential it holds for personalized medicine approaches. To this end, a comprehensive exploration of this dynamic interplay is presented, encompassing various categories of drugs, including prodrugs, antibiotics, antihypertensives, antidiabetics, anticancer agents, antipsychotics, and analgesics.

To conclude, accumulating evidence has demonstrated that the human gut microbiota can exert influence over drug metabolism, efficacy, toxicity, and inter-individual variability through the production of enzymes, metabolites, and immune modulators. The elucidation and appreciation of the extensive effects of MDI on clinical drug treatments provide a new layer of information to capture drug response and toxicity variations and thus will improve precision medicine in clinical practice. In parallel, continuing efforts were being dedicated to the development of innovative strategies to modulate gut microbiota-drug interactions, such as the utilization of probiotics, prebiotics, postbiotics, fecal microbiota transplantation, and microbial enzyme inhibitors. In summary, the interplay between human gut microbiota and pharmaceuticals is a burgeoning area of research with far-reaching implications for drug development and personalized medicine. Understanding the mechanisms of interaction and their impact on drug responses is crucial for optimizing therapeutic outcomes and advancing precision medicine, and in the next step, these research efforts will be further improved by emerging advanced measurement methods and analysis strategies such as novel sequencing technologies, standardized databases, and machine learning-driven analysis techniques.

## Figures and Tables

**Figure 1 biomedicines-12-00194-f001:**
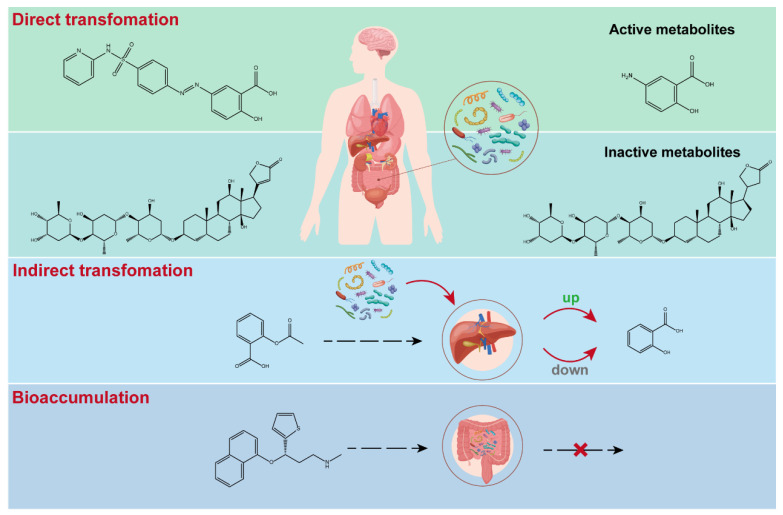
Typical paradigms of microbiota-drug interactions (MDI).

**Figure 2 biomedicines-12-00194-f002:**
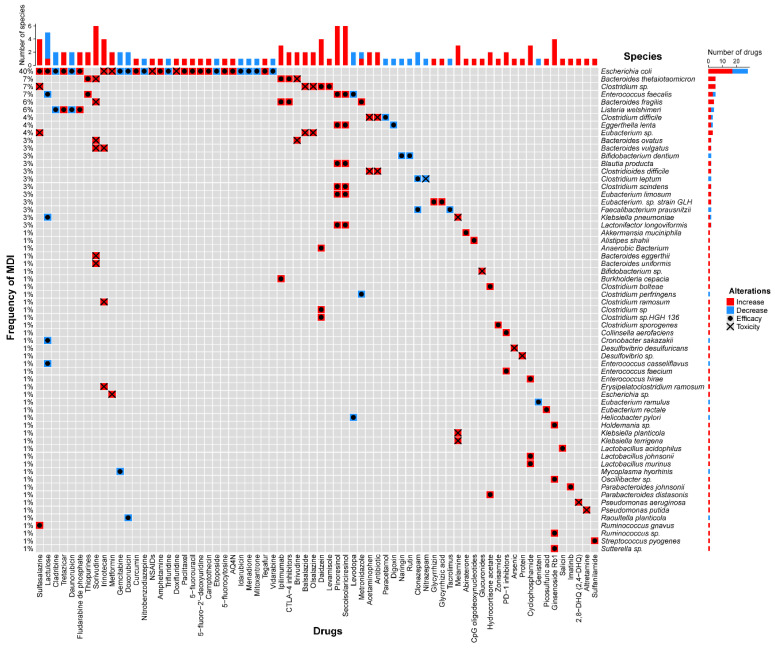
Typical examples of the impact of MDI on drug efficacy and toxicity. The diagram depicts the interactions between 63 gut microbiota species and 70 drugs (or drug categories). Each row corresponds to a specific microbiota species, while each column represents an individual drug (or drug category).

**Table 1 biomedicines-12-00194-t001:** Completed clinical trials that reported the modulation of microbiota in disease treatments.

MDI Categories	Trial Number	Conditions	Interventions	Descriptions of the Microbiota Involvement
Drugs/substances alter gut microbiota composition	NCT02086110	Autism	Synbiotic	Prebiotic and synbiotic treatments altered the stool microbiota composition of children with autism, increasing *Bifidobacterium* and reducing *Prevotella* enterotype.
	NCT02547038	*Helicobacter pylori* Infection	Pantoprazole+bismuth+tetra+metro	Reverse hybrid therapy achieved a higher eradication rate of *Helicobacter pylori* than bismuth quadruple therapy, with fewer adverse events and better compliance [78].
	NCT03018925	Ulcerative Colitis (UC)	Golimumab	Golimumab induced shifts in the abundance of bacterial markers in patients with UC, increasing *Akkermansia municiphila* and *Faecalibacterium prausnitzii* and decreasing *Escherichia coli*.
	NCT02330653	Inflammatory Bowel Diseases (IBD), Crohn Disease (CD), UC	Fecal Microbiota Transplant (FMT)	Universal donor FMT was safe and well-tolerated for pediatric patients with active UC or CD, and resulted in a significant improvement in stool microbiota composition and diversity.
	NCT02154061	Influenza	IIV flu vaccine, Metronidazole, Neomycin	IIV flu vaccine induced similar immune responses in both groups, but antibiotic treatment prior to vaccination significantly altered the gut microbiome composition and diversity, reducing *Bacteroidetes* and increasing *Firmicutes* [79].
	NCT02765256	CD	Fluconazole, Vancomycin, Neomycin, Ciprofloxacin, Prilosec, FMT	The intervention of bowel lavage and antibiotics with or without fluconazole was effective in reducing CD activity and modifying the gut microbiota.
	NCT03326583	Hyperkalemia, End Stage Renal Disease (ESRD)	Patiromer	Patiromer reduced serum potassium levels in ESRD patients with hyperkalemia, and altered gut microbiome composition, increasing *Bacteroides* and decreasing *Prevotella* [80].
	NCT01322386	Primary Sclerosing Cholangitis, Biliary Atresia	Vancomycin	Vancomycin therapy improved the liver function tests of some patients with biliary atresia or primary sclerosing cholangitis and altered the composition and diversity of their gut microbiota [81].
	NCT02328547	Irritable Bowel Syndrome (IBS)	FMT	FMT using oral capsules was safe, feasible, and tolerable for patients with diarrhea-predominant IBS, and resulted in improvement of IBS symptoms and quality of life, as well as changes in the intestinal microbiome composition and diversity of the patients [82].
	NCT02572882	End-Stage Renal Disease, Gut Microbiome Dysbiosis	Dietary Supplement: P-inulin	P-Inulin improved gut microbiome diversity and reduced inflammation in hemodialysis patients with hyperkalemia, and had no serious adverse effects [83].
	NCT01198509	Rheumatoid Arthritis, Psoriatic Arthritis, Periodontal Disease	Doxycycline, Vancomycin	Doxycycline and vancomycin had different effects on the oral and intestinal microbiota of rheumatoid arthritis patients, and that changes in the microbiota were associated with changes in disease activity and inflammatory markers [84,85].
	NCT03476317	CD	Vancomycin, Neomycin, Ciprofloxacin, FMT	The intervention of bowel lavage and antibiotics with or without fluconazole was effective in reducing CD activity and modifying the gut microbiota.
	NCT01619176	Rheumatoid Arthritis	Methotrexate, NSAID, Leflunomide Biological: Etanercept injection	Acupuncture plus conventional treatment improved the clinical symptoms and inflammatory markers of rheumatoid arthritis patients, and altered the gut microbiota composition and diversity.
	NCT02299570	Recurrent *Clostridium difficile* infection	Biological: RBX2660	RBX2660 was superior to placebo in achieving treatment success, defined as the absence of recurrent CDI or death within 8 weeks of treatment [86,87].
	NCT02646332	*Helicobacter pylori* Infection	Dexlan+amox+clar+metr, Dexlan+clarith+amox+metro	Concomitant therapy was non-inferior to reverse hybrid therapy in achieving *Helicobacter pylori* eradication. Both therapies had similar adverse event rates and patient compliance.
	NCT03181828	Urea Cycle Disorder	Acetohydroxamic acid oral tablet	Acetohydroxamic acid reduced the hydrolysis of urea by gut bacteria and increased the excretion of 13C-urea in healthy subjects and subjects with urea cycle disorders.
	NCT01839734	HIV Infection	Lubiprostone	Lubiprostone did not affect gut microbiota composition or function in HIV-infected patients with incomplete CD4^+^ T-cell recovery on antiretroviral therapy.
	NCT01355575	Non-Alcoholic Fatty Liver Disease (NAFLD)	Rifaximin	Rifaximin altered the gut microbiome composition, reduced hepatic lipid content and improved insulin sensitivity in patients with NAFLD.
Gut microbiota influences drug metabolism/disease treatment	NCT02640625	Human Immunodeficiency Virus	Probiotic compound	Probiotics were safe and well-tolerated for cART-treated INR patients with chronic HIV infection, and had a modest effect on reducing blood CD4 count and increasing blood HIV viral load [88].
	NCT02711800	Anxiety, Abdominal Pain	*Lactobacillus rhamnosus*	Probiotics reduced anxiety symptoms and abdominal pain in children with IBS, and had no serious adverse effects.
	NCT02589847	*Clostridium difficile* Infection (CDI)	Biological: RBX2660	Fecal microbiota transplantation with RBX2660 was safe and effective in preventing recurrent *Clostridium difficile* infection in patients with a history of CDI [89].
	NCT02108821	IBD, CD, UC	FMT	FMT was safe and well-tolerated, and resulted in a significant decrease in fecal calprotectin and a trend towards improvement in disease activity in pediatric patients with IBD.
	NCT02370641	Urolith	Dietary Supplement: PomX	Pomegranate extract consumption resulted in three distinct groups of urolith producers, which were associated with different gut microbiota profiles [90].
	NCT02318134	Acute Pancreatitis, Intestinal Bacteria Flora Disturbance, Intestinal Dysfunction	FMT, Normal saline	FMT improved the intestinal barrier function and reduced the systemic inflammatory response of patients with severe acute pancreatitis, and also changed the intestinal microbiome composition and diversity of the patients [91].
	NCT04322500	Chalazion	Probiotics	Probiotics improved chalaziosis resolution time and reduced recurrence rate in children compared to conservative treatment.
	NCT03061097	Antibiotic resistance	Autologous FMT	FMT was safe and feasible, and significantly reduced the prevalence and abundance of antibiotic resistant bacteria in the fecal microbiota of patients who had an infectious episode requiring antibiotics [92].
	NCT03795233	CDI	FMT, Vancomycin	FMT for primary CDI restored microbiome diversity compared to patients who did not receive FMT, and had a high rate of clinical cure and patient satisfaction.
	NCT02706717	HIV Infection	Visbiome	Visbiome reduced inflammation in HIV-infected men and women compared to placebo, and had no serious adverse effects [93].
	NCT03005379	CDI	FMT	FMT was not superior to placebo in preventing recurrent CDI in patients with a history of recurrent or severe CDI [94].
	NCT03106844	IBD, CDI	FMT	FMT was safe and effective for patients with IBD and CDI, and resulted in a significant improvement in gut microbiota composition and function.
	NCT01680640	NAFLD	Dietary Supplement: Synbiotic, Maltodextrin	Synbiotic treatment improved liver function, insulin resistance, and gut microbiota diversity in patients with NAFLD [95,96].
	NCT03621657	CDI	FMT	FMT increased the gut microbiota diversity and richness, and reduced the recurrence of *Clostridium difficile* infection in patients with a history of CDI.

**Table 2 biomedicines-12-00194-t002:** Examples of bioactive compounds derived from human gut microbiota.

Parent Compound	Metabolic Types	Active Compound	Involved Microbiota Species	Pharmacological Activity	Reference
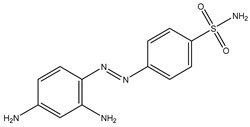	Azo reduction	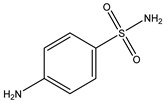	*Escherichia coli*; *Enterococcus faecalis*; *Clostridium perfringens*; *Bacteroides fragilis*	Anti-bacterial	[121]
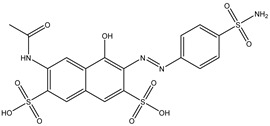	Azo reduction	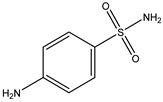	*Escherichia coli*; *Enterococcus faecalis*; *Clostridium perfringens*; *Bacteroides fragilis*	Anti-bacterial	[122]
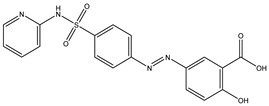	Azo reduction	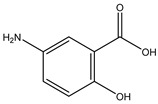	*Escherichia coli*; *Bacteroides fragilis*; *Clostridium perfringens*	Anti-bacterial	[123]
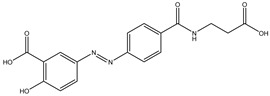	Azo reduction	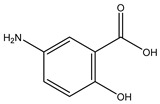	*Escherichia coli*; *Bacteroides fragilis*; *Clostridium perfringens*	Anti-bacterial	[124]
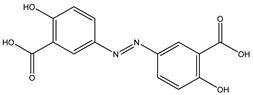	Azo reduction	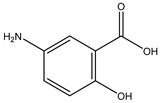	*Escherichia coli*; *Bacteroides fragilis*; *Clostridium perfringens*	Anti-bacterial	[125]
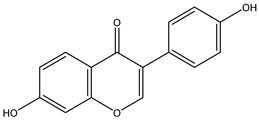	Reduction	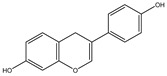	*Slackia isoflavoniconvertens*; *Lactococcus garvieae*; *Adlercreutzia equolifaciens*; *Eggerthella lenta*	Estrogenic	[126]
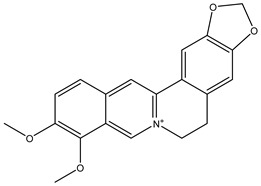	Reduction	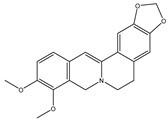	*Escherichia coli*; *Enterococcus faecalis*; *Veillonella* sp.	Anti-inflammation	[127]
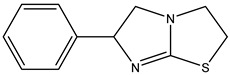	Oxidation	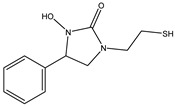	*Escherichia coli*; *Clostridium perfringens*; *Lactobacillus* sp.	Anti-neoplastic	[128]
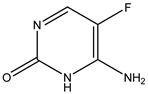	Oxidation	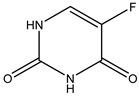	*Escherichia coli*; *Enterococcus* sp.; *Streptococcus* sp.	Anti-neoplastic	[129]
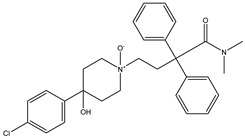	Cleavage of N-oxide bond	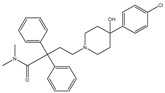	*Escherichia coli*; *Enterococcus* sp.	Anti-laxative	[130]
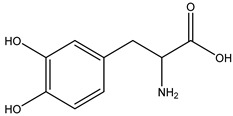	Decarboxylation	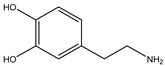	*Enterococcus* sp.; *Lactobacillus* sp.; *Streptococcus* sp.	Anti-parkinsonian	[131]
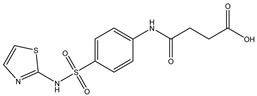	Hydrolysis	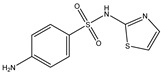	*Escherichia coli*; *Enterococcus* sp.; *Bacteroides* sp.	Anti-bacterial	[132]
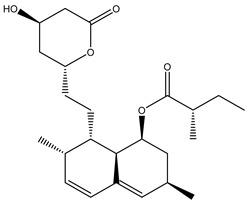	Hydrolysis	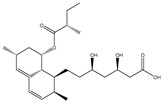	*Clostridium* sp.; *Eubacterium* sp.; *Lactobacillus* sp.	Cholesterol-lowering	[25]
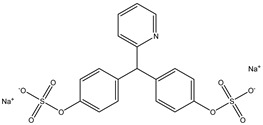	Hydrolysis	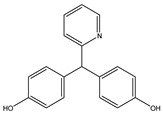	*Escherichia coli*; *Bacteroides fragilis*; *Clostridium perfringens*	Laxative	[133]
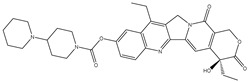	Hydrolysis	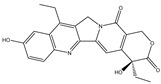	*Escherichia coli*; *Bifidobacterium longum*; *Enterococcus faecalis*	Anti-neoplastic	[7]
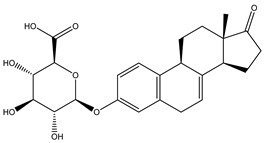	Deconjugation	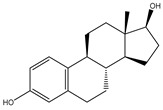	*Escherichia coli*; *Eubacterium limosum*; *Ruminococcus* sp.	Estrogenic	[31]
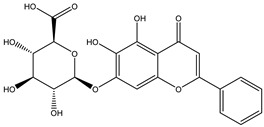	Deconjugation	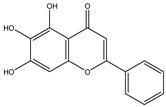	*Escherichia coli*; *Bacteroides fragilis*; *Eubacterium ramulus*; *Firmicutes* sp.	Anti-inflammatory	[134]
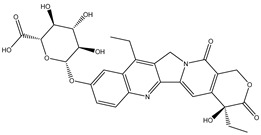	Deconjugation	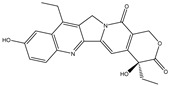	*Escherichia coli*; *Clostridium* sp.; *Ruminococcus* sp.; *Bifidobacterium* sp.	Anti-neoplastic	[135]

## Data Availability

This is a review article, therefore all data supporting reported results are published and cited in the bibliography.

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
