# Peer review of "Mechanisms and Clinical Implications of Human Gut Microbiota-Drug Interactions in the Precision Medicine Era"

_biomedicines, 2024, doi:10.3390/biomedicines12010194_

Round 1

Reviewer 1 Report

Comments and Suggestions for Authors

P. 1; line 13: What is represented by the abbreviation “MDI”? It is not making sense as written.

P. 1; line 32: Something missing after “underpinning”, so either add the missing part or delete “and” coming after “underpinning”.

Line 78: The word “host” should be moved before “metabolizing enzyme”.

Line 151: The sentence about tetracycline should be re-written because the present form is not easy to understand.

Line 215: Something must be written after “gut” in “can traverse from the gut” to make it sense.

Lines 219-222: These lines should be deleted because they are not related with immune function.

Comments on the Quality of English Language

English is acceptable; only some minor editing is needed.

Author Response

Dear reviewer,

Thank you for your review and valuable comments. We appreciate your recognition and affirmation of our work, as well as your suggestions and questions. We have made corresponding modifications and additions to the manuscript to address these comments. Here are the main points of how we have revised our manuscript:

In response to revision comment 1, we have highlighted the initial letters of "MDI" in uppercase.

In response to revision comment 2, we have removed the conjunction "and".

In response to revision comment 3, we have rearranged the order of "host" to "drug-host metabolizing enzymes."

In response to revision comment 4, we have provided additional clarification on examples of tetracycline affecting antibiotic resistance of gut microbiota.

In response to revision comment 5, we have supplemented explanations on the impact of small molecule metabolites on the immune system, to make it more logical.

In response to revision comment 6, we have eliminated paragraphs unrelated to the topic and added a section introducing the interaction between gut microbiota and immunosuppressive drugs.

Additionally, we have rectified some other grammar mistakes and typos in the manuscript, with all modifications highlighted in red color. We hope that our modifications can meet your expectations. If there are any questions or comments, please feel free to contact us. Thank you again for your review comments and suggestions. We look forward to hearing from you.

Reviewer 2 Report

Comments and Suggestions for Authors

The human microbiome comprises trillions of bacteria, archaea, viruses, protozoans and fungi that form the symbiotic microbial cells hosted by each individual in various sites (e.g. oral cavity, skin, gastrointestinal and urogenital tract). The gut microbiome, which is the most important in terms of density and diversity, plays a key role in protecting against pathogens, maintaining the integrity of the digestive epithelial barrier and contributing to digestion and metabolism. Several factors can modulate the composition of the gut microbiome (e.g., aging, diet or medication). Imbalance of the gut microbiome, also known as dysbiosis, has been associated with epithelial barrier dysfunction and local or systemic disorders, including activation of the immune system. Therefore, topic regarding gut microbiota as a modulator of therapeutic of therapeutic response is very important and current.

The aim of the presented study in manuscript is mechanism and clinical implications of human gut microbiota-drug interactions. The review also discussed the therapeutic potential of modulating the gut microbiota to optimize clinical treatments and personalized medicine. Molecular mechanisms of microbiota-drug interactions were presented in details by the authors. The authors emphasized the role of the gut microbiota in drug metabolism by influencing the enterohepatic circulation of drugs. In this process, the gut microbiota plays a crucial role in bile acid metabolism.

A perfect example to illustrate this interaction between the immunosuppressive drugs and the gut microbiome is the case of mycophenolic acid, active metabolite of mycophenolate mofetil. This pharmacological profile may explain its ability to modify the microbial composition and metabolism. It is a pity that the authors did not include such an important group of drugs in their work. Immunosuppressive drugs could alter the composition of the gut microbiome, which could influence the metabolism of immunosuppressive drugs and the immune system of transplant patients. The gut microbiome offers a new opportunities for the precision medicine in transplantation.

The manuscript is well-written, the figures are legible and appropriate to the manuscript. However, my major concern is focused on the table presenting clinical trials for evaluating the modulation of microbiota in disease treatments. In my opinion, it should be modified and it should be arranged according to the selected key. Then, they will be more readable.  

Author Response

Dear reviewer:

Thank you for your review and valuable comments. We appreciate your recognition and affirmation of our work, as well as your suggestions and questions. We have made corresponding modifications and additions to the manuscript to address these comments. Here are the point-by-point responses to your comments and details of how we have revised our manuscript:

Response to comment 1:

Thank you for highlighting the noteworthy interaction between immunosuppressive drugs and the gut microbiota. We appreciate your insightful suggestion, and indeed, this significant aspect was unintentionally omitted from our previous manuscript. In the revised manuscript, we have incorporated a dedicated subsection within the Immune Modulation section. This new section introduces mycophenolic acid as a typical example and delves into the intricate interplay between immunosuppressive drugs and the gut microbiota. Additionally, it highlights the potential applications of precision medicine in the field of transplantation. (See line 220-233 for details)

Response to comment 2:

Your suggestion to improve the readability of the table by using selected key is very helpful. We have classified the clinical trials into two groups by their relevance to MDI: “Drugs/substances alter gut microbiota composition” and “Gut microbiota influences drug metabolism/disease treatment”. We have also deleted a few unclassified clinical trials to make the information in the table more focused, and improved the format of the table to make it clearer and more readable. (See Table 1 for details)

The above-mentioned revisions have been marked in red color, and we have also corrected some grammar mistakes and typos in the manuscript. We hope that our modifications can meet your expectations. If there are any questions or comments, please feel free to contact us. Thank you again for your review comments and suggestions. We look forward to hearing from you.
